# Neutralizing-antibody response to SARS-CoV-2 for 12 months after the COVID-19 workplace outbreaks in Japan

Azumi Ishizaki[1], Xiuqiong Bi[1], Quynh Thi Nguyen[1], Tomomi Maeno[2], Akinori Hara[3,4], Hiroyuki Nakamura[3,4], Sanae Kuramoto[5], Koichi Nishi[6], Hiroyasu Ooe[7], Hiroshi Ichimura[1]*

1 Department of Viral Infection and International Health, Graduate School of Medical Science, Kanazawa University, Kanazawa, Japan, 2 Kaga Toshiba Electronics, Ishikawa, Japan, 3 Nexco, Ishikawa, Japan, 4 Department of Public Health, Graduate School of Medical Science, Kanazawa University, Kanazawa, Japan, 5 Ishikawa Prefectural Institute of Public Health and Environmental Science, Kanazawa, Japan, 6 Ishikawa Prefectural Central Hospital, Kanazawa, Japan, 7 Department of Laboratory, Kanazawa University Hospital, Kanazawa, Japan

* ichimura@med.kanazawa-u.ac.jp

**Data Availability Statement:** All relevant data are within the manuscript and its Supporting Information files.

## Abstract

This study aimed to elucidate the 12-month durability of neutralizing antibodies (NAbs) against severe acute respiratory syndrome coronavirus 2 (SARS-CoV-2) in patients infected during the 2020 workplace outbreaks of coronavirus disease 2019 (COVID-19) in Japan. We followed 33 Japanese patients infected with SARS-CoV-2 in April 2020 for 12 months (12M). Patients were tested for NAbs and for antibodies against the SARS-CoV-2 nucleocapsid (anti-NC-Ab) and antibodies against the spike receptor-binding domain (anti-RBD-Ab). Tests were performed at 2M, 6M, and 12M after the primary infection (*api*) with commercially available test kits. In 90.9% (30/33) of patients, NAbs persisted for 12M *api*, though the median titers significantly declined from 78.7% (interquartile range [IQR]: 73.0–85.0%) at 2M, to 59.8% (IQR: 51.2–77.9) at 6M ($P = 0.008$), and to 56.2% (IQR: 39.6–74.4) at 12M ($P<0.001$). An exponential decay model showed that the NAb level reached undetectable concentrations at 35.5 months *api* (95% confidence interval: 26.5–48.0 months). Additionally, NAb titers were significantly related to anti-RBD-Ab titers (rho = 0.736, $P<0.001$), but not to anti-NC-Ab titers. In most patients convalescing from COVID-19, NAbs persisted for 12M *api*. This result suggested that patients need a booster vaccination within one year *api*, even though NAbs could be detected for over two years *api*. Anti-RBD-Ab titers could be used as a surrogate marker for predicting residual NAb levels.

## Introduction

Coronavirus disease 2019 (COVID-19), caused by severe acute respiratory syndrome coronavirus 2 (SARS-CoV-2), emerged in December 2019, and has caused a global pandemic [1]. Humoral immune responses, particularly antibodies specific to the receptor-binding domain (RBD) in the spike protein of SARS-CoV-2, play an important role in viral neutralization and

**Funding:** The author(s) received no specific funding for this work.

**Competing interests:** The authors have declared that no competing interests exist.

clearance [2]. Several longitudinal cohort studies of patients with COVID-19 from China, the US, and Thailand have shown that neutralizing antibody (NAb) titers reached a peak around one month post-symptom onset, then declined, but they remained detectable for 12–13 months (12–13 M) post-symptom onset in over 95% of patients convalescing from COVID-19 [3–6].

Previous studies of four seasonal coronaviruses showed that humoral immunities against the viruses were short-lasting, and re-infections by the same coronavirus could occur within an interval of 12 M [7, 8]. SARS-CoV-2 re-infections were reported in 11%–16% of patients convalescing from COVID-19 within 6 M after the primary infection in large, population-based studies in the UK and US [9, 10]. To understand the long-term kinetics of humoral immune responses to primary SARS-CoV-2 infections, it is essential to evaluate the risk of re-infection and determine the appropriate timing for booster vaccinations, if necessary, to prevent re-infections.

This study aimed to elucidate the durability of antibodies against SARS-CoV-2, including antibodies against the nucleocapsid protein (anti-NC-Ab), antibodies against the RBD in the spike protein (anti-RBD-Ab), and neutralizing antibodies (NAbs). The test cohort included individuals infected with SARS-CoV-2 during the COVID-19 workplace outbreaks, which occurred in Ishikawa prefecture, Japan, in April 2020.

## Methods

### Study population and schedule for specimen collections

We recruited employees that acquired confirmed SARS-CoV-2 infections during the two outbreaks of COVID-19, which occurred at two workplaces in Ishikawa, Japan, in April 2020 (index cases). SARS-CoV-2 infections were confirmed with reverse transcription polymerase chain reaction (RT-PCR) tests for detecting viral RNA. We also recruited "close contacts", defined as individuals who had been in close contact with an index case without taking necessary infection prevention measures, such as wearing a mask, practicing social distancing, and so on. Close contacts were identified by the contact-tracing surveillance team of the local government in Ishikawa Prefecture, Japan, and included all household members of the index cases. Close contacts underwent RT-PCR tests for SARS-CoV-2, based on the circumstances of their contact with the index case(s) and their clinical symptoms [11]. All household members of the index case underwent RT-PCR tests.

Blood specimens were collected from index patients between June and July 2020, 2–3 M after the COVID-19 outbreak in the workplace. Individuals that showed positive anti-NC-Ab results at 2 M were followed at 6 M (i.e., in November and December 2020) and at 12 M (i.e., in April 2021) after the workplace outbreaks.

### Laboratory procedures

After blood collection, sera were separated by centrifugation at 3000 × g for 10 min and stored at −80˚C until use. Anti-NC-Ab and anti-RBD-Ab were quantified with electrochemiluminescence immunoassays, Elecsys® Anti-SARS-CoV-2 (S300) RUO and Elecsys® Anti-SARS--CoV-2 S (S300) RUO (Roche Diagnostics, IN, USA), respectively. These assays were performed with a fully automated Cobas 8000 Analyzer Series Module e801 (Roche Diagnostics). Results are expressed in terms of a cutoff index (COI) or a cutoff concentration. The presence of anti-NC-Ab was positive when the detection value was ≥1.0 COI, and the presence of anti-RBD-Ab was positive when the detection value was ≥0.8 U/mL. The ability of NAbs to inhibit SARS-CoV-2 binding to its target was quantified, in duplicate, with an enzyme-linked immunosorbent assay (ELISA)-based SARS-CoV-2 Surrogate Virus neutralization Test

(sVNT) Kit (GenScript, Jiangsu, China) [12]. The cutoff level (lower limit of quantification) for the positive detection of NAbs was set at 30% inhibition, according to manufacturer instructions.

## Statistical analysis

Statistical analyses were performed with SPSS version 25 (SPSS Japan Inc., Tokyo, Japan). The chi-squared test or Fisher's exact test and Mann-Whitney $U$ test were performed to assess differences between groups. Changes in antibody levels were compared between paired samples with the Friedman test, and the Bonferroni correction was applied in post hoc analyses. Spearman's rank correlation coefficient was used to assess correlations between the levels of anti-NC-Ab, anti-RBD-Ab, and NAbs. A receiver operating characteristic (ROC) curve was plotted to evaluate the sensitivity and specificity of anti-RBD-Abs to predict the NAb level. Simple linear regression analysis was conducted to test associations between the levels of antibodies at 2 M, 6 M, and 12 M, and relevant variables, including age, sex, comorbidity, and disease severity. The NAb decay rate was calculated with an exponential decay model [13]. $P$-values $<0.05$ were considered significant.

## Ethical approval

This study was carried out according to the World Medical Association's Declaration of Helsinki and the Japanese Ethics Guidelines for Human Genome/Gene Analysis Research. The study protocol was approved by the ethics committee of Kanazawa University (3388–1). All participants were appropriately informed about the study, and all provided written consent to participate in this study. When a participant under 18 years old agreed to participate, consent was obtained from the parents or guardians.

# Results

## Study participants

We recruited 135 company employees, including 27 index cases (21 males and 6 females; median age: 48.0 years, interquartile range [IQR]: 41.0–54.0; Table 1) and 108 close contacts (86 males and 22 females; median age: 49.0 years, IQR: 42.3–54.8). The sex and age distributions of participants did not significantly differ between the index cases and close contacts ($P = 0.832$ and $P = 0.684$, respectively). We also included 11 household members of the index cases (2 males and 9 females; median age: 46.0 years, IQR: 14.0–67.0). None of the participants had received the COVID-19 vaccine or exhibited any clinical symptom suggestive of COVID-19 during the follow-up period.

**Table 1. Gender and age distribution of the participants and their SARS-CoV-2 PCR and anti-NC-Ab results at 2M.**

| | N | Sex Male (%) | Age Median, IQR (years) | Anti-NC-Ab positive at 2M, n (%) |
|---|---|---|---|---|
| Index cases (RT-PCR positive) | 27 | 21 (77.8) | 48.0 (41.0–54.0) | 27 (100%) |
| Closed contact colleagues (RT-PCR negative or not tested) | 108 | 86 (79.6)[1] | 49.0 (42.3–54.8)[2] | 5 (4.6%) |
| Close contact household members | 11 | 2 (18.2) | 46.0 (14.0–67.0) | 3 (27.3%) |
| RT-PCR positive | 3 | 1 (33.3) | 46.0 (11.0–) | 3 (100%) |
| RT-PCR negative | 8 | 1 (12.5) | 49.5 (16.0–74.5) | 0 (0%) |

RT-PCR: reverse transcription polymerase chain reaction for severe acute respiratory syndrome coronavirus 2 (SARS-CoV-2); IQR: interquartile range; Anti-NC-Ab: antibody to the nucleocapsid of SARS-CoV-2; 2M: 2 to 3 months after the COVID-19 outbreak in the workplace; $P$-values between index cases versus close contacts are based on the Mann-Whitney $U$ test or Fisher's exact test; 1: $P = 0.832$; 2: $P = 0.684$.

## Antibodies at 2–3 months after the COVID-19 workplace outbreaks

All 27 index cases showed positive results for the anti-NC-Ab. Among the 108 close contacts, five (4.6%) showed positive results for the anti-NC-Ab (Table 1). These five close contacts had not undergone RT-PCR testing during the workplace outbreaks. Of the 11 household members, three (27.3%) had sustained SARS-CoV-2 infections, confirmed by RT-PCR, and all three showed positive anti-NC-Ab results. Thus, a total of 35 participants had been infected with SARS-CoV-2 during the outbreaks. Additionally, all of these 35 participants showed positive anti-RBD-Ab and NAb results.

## Changes in anti-NC-Ab levels at 6 and 12 months after the outbreaks

Among the 35 individuals with positive anti-NC-Ab results, 33 agreed to additional follow-ups at 6 and 12 M. Of the 33, three were asymptomatic, 27 had mild symptoms, and the remaining three had moderate symptoms. Among these 33 individuals, 33 (100%) and 32 (97.0%) showed positive anti-NC-Ab results at 6 and 12 M, respectively (S1 Table).

The anti-NC-Ab titers declined significantly from a median COI of 114.8 (IQR: 65.4–188.0) at 2 M, to a median COI of 69.4 (IQR: 35.0–115.8) at 6 M and a median COI of 29.5 (IQR: 12.9–49.8) at 12 M (2M vs. 6M, $P<0.001$; 2 M vs. 12 M, $P<0.001$; 6 M vs. 12 M, $P = 0.001$; Fig 1A).

## Changes in anti-RBD-Ab levels at 6 and 12 months after the outbreaks

All 33 participants included in 12-M follow-ups showed positive anti-RBD-Ab results at both 6 and 12 M. The anti-RBD-Ab titers did not significantly change over time. The median anti-

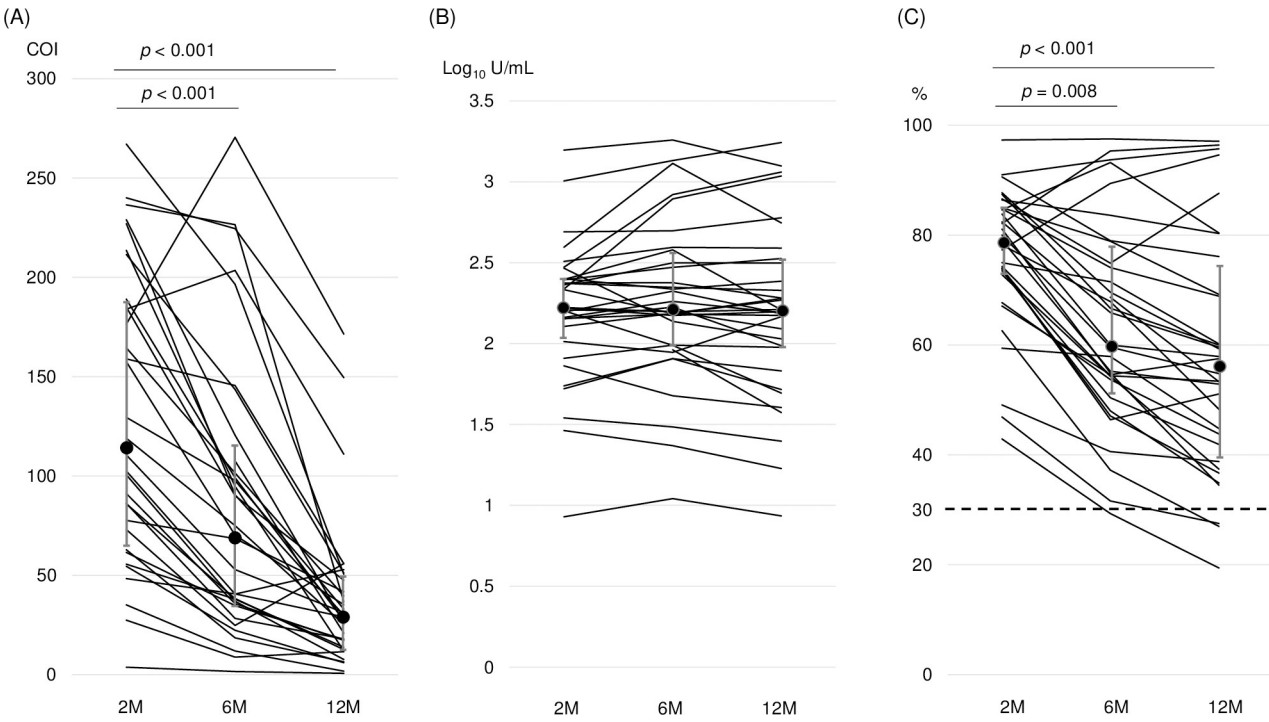

**Fig 1. The changes of antibody responses from 2M, 6M, to 12M after the outbreaks of COVID-19 at workplaces among 33 with asymptomatic, mild to moderate COVID-19.** (A) The changes of the anti-nucleocapsid antibody. Cutoff index (COI) $\geqq 1.0$ is positive. (B) The changes of the anti-receptor-binding domain antibody. The numeric value $\geq 0.8$ U/mL [$Log_{10}(0.8) = -0.0969$] is positive. (C) The changes of the inhibition rate of antibody specific to the receptor-binding domain that is equivalent to neutralizing antibody. The positive cutoff is $\geqq 30\%$, indicated as the dotted line. Significance among three points was analyzed by the Friedman test and post hoc analysis with a Bonferroni correction. The dot and error bars correspond to the median for anti-nucleocapsid antibody and anti-receptor-binding domain and the median titer for inhibition rate of neutralizing antibody ± interquartile range.

RBD-Ab titers were ($\log_{10}$ U/mL) 2.2 (IQR: 2.0–2.4) at 2 M, 2.2 (IQR: 2.0–2.6) at 6 M, and 2.2 (IQR: 2.0–2.5) at 12 M ($P = 0.14$ among the three groups; Fig 1B).

## Changes in NAb levels at 6 and 12 months after the outbreaks

Of the 33 participants followed, 32 (97.0%) and 30 (90.9%) showed positive NAb results at 6 and 12 M, respectively. The median NAb titer significantly declined over time, with 78.7% inhibition (IQR: 73.0–85.0%) at 2 M, 59.8% (IQR: 51.2–77.9) at 6 M ($P = 0.008$), and 56.2% (IQR: 39.6–74.4) at 12 M (2 M vs. 12 M, $P<0.001$; 6 M vs. 12 M, $P = 0.026$; Fig 1C).

In three individuals, NAb titer levels declined from 2 to 6 M, then rebounded at 12 M. In the other three individuals, NAb titer levels increased steadily, from 2 to 12 M. However, in these six participants, the anti-NC-Ab titer levels decreased constantly over the same observation periods (Fig 1B and Table 2). In one participant, the anti-NC-Ab result was negative at 12 M, but the anti-RBD-Ab and NAb results remained positive at 12 M (Table 2).

## Correlation between the NAb and anti-RBD-Ab titers

NAb titers were strongly correlated with the anti-RBD-Ab titers at all time points (rho = 0.739 at 2 M; rho = 0.707 at 6 M; rho = 0.801 at 12 M; all $P<0.001$) and over the entire study period (rho = 0.736, $P<0.001$; Fig 2A). The ROC curve analysis indicated an anti-RBD-Ab cutoff threshold of 1.3 $\log_{10}$ U/mL for the ability of anti-RBD-Ab to predict NAb positivity, with a sensitivity of 98.9% and a specificity of 75.0%.

The NAb titers were not significantly correlated with anti-NC-Ab titers at any time point (rho = 0.062 at 2 M, $P = 0.73$; rho = 0.129 at 6 M, $P = 0.48$; and rho = 0.268 at 12 M, $P = 0.13$). However, over the entire study period, NAb titers were mildly correlated with anti-NC-Ab titers (rho = 0.355; $P<0.001$; Fig 2B). Other factors, including age, sex, disease severity, and comorbidity, were not significantly associated with the titers of anti-NC-Ab, anti-RBD-Ab, or NAb, at any time point ($P>0.05$ for all analyses, S2–S4 Tables).

## Prediction of NAb durability

We estimated that the NAb level declined to undetectable values at about 35.5 months (95% confidence interval [CI]: 26.5–48.0 months) after the outbreak (Fig 3).

**Table 2. The changes of the anti-NC-Ab, anti-RBD-Ab, and NAb among the individuals whose NAb titer increased from 6M to 12M and whose anti-NC-Ab turned to be negative.**

| ID | Sex | Age | Comorbidity | Severity | Anti-NC-Ab (COI) | | | Anti-RBD-Ab ($\log_{10}$ U/mL) | | | NAb (%) | | |
|---|---|---|---|---|---|---|---|---|---|---|---|---|---|
| | | | | | 2M | 6M | 12M | 2M | 6M | 12M | 2M | 6M | 12M |
| NAb re-increased at 12M than 6M | | | | | | | | | | | | | |
| 24 | M | 38 | No | Mild | 267 | 196.5 | 50.6 | 2.01 | 1.94 | 2.17 | 74 | 46.4 | 51.1 |
| 25 | M | 49 | No | Mild | 110.5 | 52.9 | 31.2 | 2.11 | 2.19 | 2.28 | 67.1 | 54.6 | 57.5 |
| 26 | M | 11 | No | Mild | 164.5 | 101.5 | 39.6 | 2.69 | 2.70 | 2.78 | 86.6 | 74.9 | 87.6 |
| NAb increased in 6-12M than 2M | | | | | | | | | | | | | |
| 27 | M | 56 | Yes | Mild | 91.1 | 35.7 | 17.2 | 2.34 | 2.89 | 3.04 | 82.1 | 95.3 | 96.4 |
| 28 | F | 32 | No | Mild | 61.5 | 34.7 | 17.7 | 2.47 | 2.92 | 3.06 | 77.3 | 89.4 | 94.6 |
| 29 | M | 39 | No | Mild | 240 | 224.5 | 149.5 | 3.01 | 3.13 | 3.24 | 91 | 93.7 | 95.7 |
| Anti-NC-Ab became negative | | | | | | | | | | | | | |
| 30 | M | 52 | No | Mild | 3.7 | 1.5 | 0.6 | 1.91 | 1.99 | 1.98 | 49.1 | 40.6 | 38.8 |

Anti-NC-Ab: anti-nucleocapsid antibody, $\geq 1.0$ COI (cut-off index) is positive; Anti-RBD-Ab: anti-receptor binding domain antibody, $\geq -0.0969$ $\log_{10}$ U/mL ($\geq 0.8$ U/ml) is positive; NAb: neutralizing antibody, $\geq 30\%$ (% of inhibition) is positive; 2M: 2 to 3 months after the COVID-19 outbreak in the workplace; 6M: 6 months after the outbreak; 12M: 12 months after the outbreak; Mild: no pneumonia; Moderate: pneumonia not requiring oxygen.

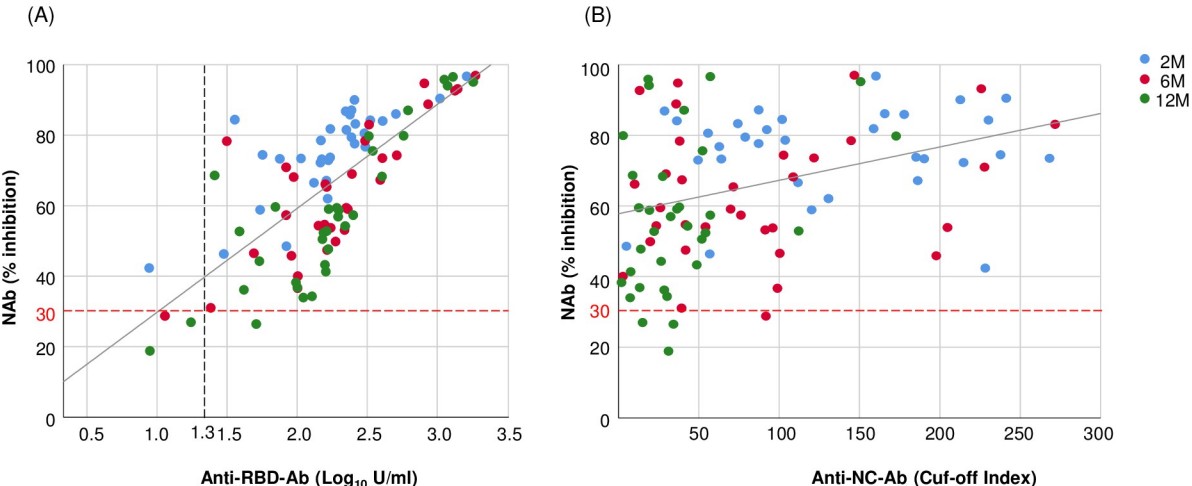

**Fig 2. Correlations of the neutralizing antibody (NAb) with anti-RBD-Ab and anti-NC-Ab.** (A) Correlation between the NAb and the anti-receptor-binding domain antibody (anti-RBD-Ab). The cutoff for anti-RBD-Ab, $Log_{10}$ (0.8) = −0.0969, is not shown in the graph. Spearman's rank correlation coefficient: 2M: rho = 0.739; 6M: rho = 0.707; 12M: rho = 0.801; in total: rho = 0.736; all $P < 0.001$. Blue dots: data at 2 months after the outbreaks of COVID-19 at workplaces (2M); red dots: data at 6 months (6M); and green dots: data at 12 months (12M). The solid line represents the linear regression. The red dotted line indicates the cutoff of NAb. The black dotted line indicates the suggested threshold of anti-RBD-Ab to determine the positivity of NAb. (B) Correlation between the NAb and the anti-nucleocapsid antibody (anti-NC-Ab). The cutoff for anti-NC-Ab is Cut off index (COI) $\geqq$ 1.0. Spearman's rank correlation coefficient: 2M: rho = 0.062, $p = 0.73$; 6M: rho = 0.129, $p = 0.48$; 12M: rho = 0.268, $p = 0.13$; in total: rho = 0.355, $P < 0.001$.

## Discussion

In this study, we analyzed the 12-month durability of NAb titers in 33 individuals with first-time SARS-CoV-2 infections that displayed zero to moderate symptoms in Japan. We found

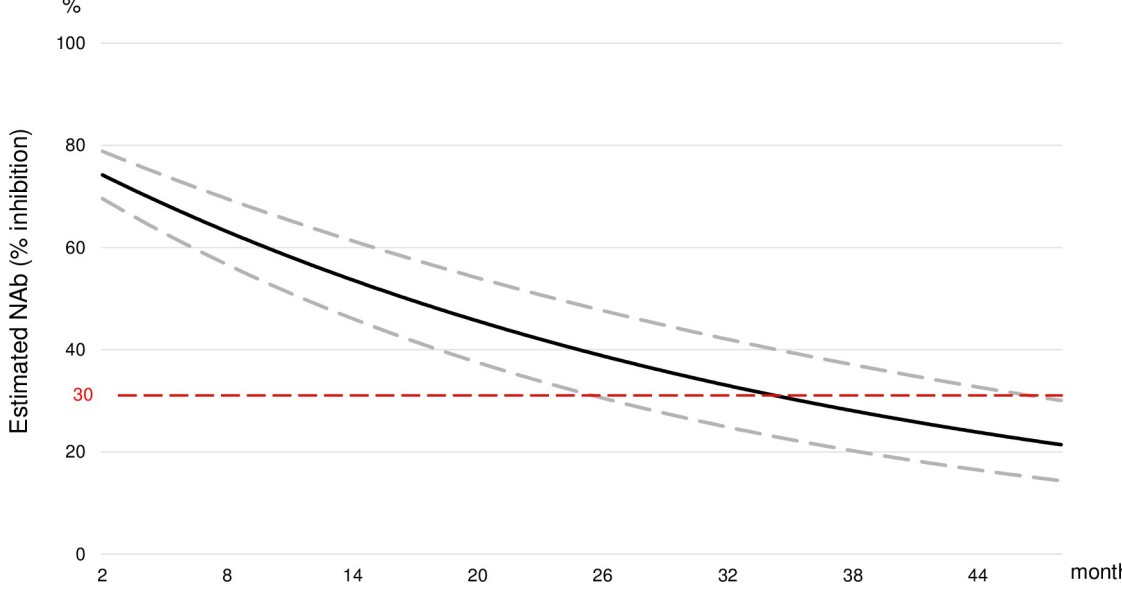

**Fig 3. The estimated decay curve of best fit for neutralizing antibody (NAb).** The solid line indicates the estimated level of the NAb and dotted lines indicate the upper and the lower bounds of the 95% confidence intervals. The red dotted line indicates the cutoff line for 30% of NAb.

that 90.9% (30/33) of these individuals maintained NAbs for 12 months, although NAb titers had declined significantly at 12 M, regardless of age, sex, comorbidity, and disease severity.

Plasma NAb levels are known to be correlated with the prevention of SARS-CoV-2 infections and COVID-19 onset, and/or the reduction of disease severity. However, to date, no consensus has been established on the specific titer needed to prevent infection [14]. Townsend et al. estimated that, after a primary infection, the likely time interval for a SARS-CoV-2 reinfection was between 3 months and 5.1 years, with a median of 16 months [15]. Nguyen et al. reported that the SARS-CoV-2 delta variant was capable of breakthrough infections among fully vaccinated Vietnamese health care workers. In those workers, the median NAb titer determined with the sVNT was 59.4% (IQR: 32.5–73.1%) at diagnosis [16]. Our exponential decay model showed that the NAb level would decline to undetectable levels (30% inhibition) at 35.5 months (95% CI: 26.5–48) after a SARS-CoV-2 infection, and that NAb levels would fall to 59.4% inhibition at 10.3 months (95% CI: 6.6–15.5) after the primary infection. Because a decline in NAbs increases the susceptibility to a breakthrough infection [17], our findings suggested that individuals convalescing from COVID-19 should receive a booster vaccination within one year after the primary infection to prevent re-infections [5, 18–20].

Our results also showed that NAb titers at 2, 6, and 12 M did not correlate with disease severity. Several previous studies reported that the magnitude of the peak NAb titer depended on disease severity during early convalescence [6, 21–23]. However, studies on the differences between severity groups in NAb titers measured 3 to 12 M after symptom onset have produced discrepant results [4, 6, 22, 23]. These discrepancies could be due to differences in the target populations. For example, our study did not include severe cases, but other studies included severe and critical cases. Alternatively, the discrepancies could be due to differences in the methods for detecting NAbs. In the present study, we used the sVNT, but other studies used neutralizing assays with pseudotyped virus and/or authentic virus. Importantly, all four longitudinal cohort studies with observation periods of 12–13 M showed that NAb titers declined over time [3–6]. Thus, booster vaccinations should be recommended for all patients convalescing from COVID-19, regardless of disease severity.

We found that NAb titers were strongly correlated with anti-RBD-Ab titers at each time point and over the entire study period (all rho>0.7; $P<0.001$). Our ROC curve analysis indicated that anti-RBD-Ab titers $\geq 1.3$ $Log_{10}$ U/mL (equivalent to $\geq 20$ U/ml), could predict NAb positivity with a sensitivity of 98.9% and specificity of 75.0%. The anti-RBD-Ab assay utilized in the current study was fully automated, and the run-time was short, compared to the run-time for the manual sVNT assay. Considering these findings, the anti-RBD-Ab titer could be a useful alternative marker for predicting NAb positivity and for monitoring the level of herd immunity in a community.

In six participants of our cohort, the anti-RBD-Ab and NAb levels increased from 2 M to 6 and 12 M. However, the anti-NC-Ab titers continued to decline over the same period. None of these participants reported clinical COVID-19 symptoms during the study period. Some previous longitudinal studies also reported an increase in NAb in patients convalescing from COVID-19 during 6 to 12 M after disease onset, but no mechanism has been proposed [4, 24]. We speculate that this continuous humoral response after the convalescent period could be induced by re-exposure to SARS-CoV-2, by the persistence of spike proteins, or by cross-reactivity to other human seasonal coronaviruses. In the current study, several SARS-CoV-2 endemics had occurred in the area during the study period. Thus, a re-exposure to SARS-CoV-2 without triggering an infection might have boosted the anti-RBD-Ab and NAb titers without increasing the anti-NC-Ab titer in six individuals in our study. Additionally, Gaebler et al. reported that persistent SARS-CoV-2 antigens were detected in individuals convalescing from COVID-19; viral antigen detection persisted for an average of 4 M after disease onset

[25]. Additionally, antibodies against the S proteins of seasonal human coronavirus were shown to be boosted by a SARS-CoV-2 infection [26, 27]. Therefore, the anti-SARS-CoV-2 antibodies might have been induced by cross-reactivity to other seasonal human coronavirus infections [28]. In this study, we did not conduct RT-PCR tests to detect SARS-CoV-2 RNA during observation period, since none of the participants exhibited any clinical symptom suggestive of COVID-19 and testing capacity for COVD-19 RT-PCR in Japan was limited during the study period, nor did we investigate viral antigens or antibodies to other human coronaviruses in individuals convalescing from COVID-19. Further investigations are needed to understand the underlying mechanisms of prolonged humoral responses in patients convalescing from COVID-19.

The present study showed that, during the COVID-19 outbreaks at workplaces, 4.6% of close contacts were missed in diagnosing COVID-19. To control the epidemic in Japan, more active surveillance is needed to identify all COVID-19 cases, including asymptomatic cases. Moreover, we observed secondary attacks in 27.3% of households; this rate was slightly higher than those reported previously (6 to 16.6%) in systematic reviews on household SARS-CoV-2 transmission [29, 30]. The Japanese national whole-genome epidemiologic surveillance project demonstrated that the SARS-CoV-2 strain B.1.1. variant caused the epidemic that occurred in late March through April 2020 in Japan, including the area sampled in the present study [31]. In contrast, the omicron variants were the predominant circulating strain in the latest epidemic in Japan. The omicron variant showed higher transmissibility within households than previously circulating variants [32]. Thus, more caution and stronger measures to prevent household transmission are needed in Japan.

The current study has several limitations. First, the small number of COVID-19 cases included could have limited the significance of the findings. Second, none of the study participants had the opportunity to receive COVID-19 vaccination during the observation period, from June 2020 to April 2021, since the COVID-19 vaccination program for the general population in Japan began in June 2021. Recently, however, it was reported that 58 individuals with SARS-CoV-2 infection history in the same area of the current study, including 19 of the current study participants, had a median anti-RBD-Ab titer of 4.2 $\log_{10}$ U/mL [range 2.8 to 5.0] and NAbs titers of more than 95% after two doses of the vaccines [33]. This recent report would support our suggestion that a booster vaccination within one year after the primary infection will be beneficial, considering a median anti-RBD-Ab titer of 2.2 $\log_{10}$ U/mL (IQR: 2.0–2.5) and a median NAbs titer of 56.2% (IQR: 39.6–74.4) at 12M in the current study. Third, we utilized a surrogate virus neutralization test. This ELISA was based on antibody-mediated blockage of the interaction between the angiotensin-converting enzyme 2 receptor protein and the viral RBD [12]. However, a titration of the neutralizing antibody against an authentic virus requires biosafety level (BSL) 3 facilities. Alternatively, the sVNT assay was reported to be comparable with the virus neutralization test against an authentic virus and/or pseudo-type virus [12]. Thus, we adopted the sVNT, which is simple and can be performed in a BSL2 laboratory.

Currently, infections by new variants in convalescing or vaccinated individuals is a growing concern for COVID-19 containment [34–36]. Therefore, in future, we must also evaluate the sera in participants that were infected with the B.1.1. variant to determine the neutralizing capacity against emerging variants. That evaluation will require an authentic or pseudo-type virus-based assay.

## Conclusion

Our findings suggested that patients convalescing from a primary SARS-CoV-2 infection should receive a COVID-19 booster vaccination within one year after the infection. We also

showed that the anti-RBD-Ab could be used as a surrogate marker to predict the residual level of NAbs. Our findings provided new insight into the durability of humoral immune responses to SARS-CoV-2 and suggested an appropriate timing for booster vaccinations in patients convalescing from COVID-19.

## Supporting information

**S1 Table. Details of the antibody titers.**
(DOCX)

**S2 Table. Simple linear regression analysis of anti-NC-Ab with age, sex, comorbidity, and disease severity at 2M, 6M, and 12M.**
(DOCX)

**S3 Table. Simple linear regression analysis of anti-RBD-Ab with age, sex, comorbidity, and disease severity at 2M, 6M, and 12M.**
(DOCX)

**S4 Table. Simple linear regression analysis of NAb with age, sex, comorbidity, and disease severity at 2M, 6M, and 12M.**
(DOCX)

## Acknowledgments

We would like to acknowledge the participants of this study. We would also like to thank Ms. Michiko Yamamoto and Ms. Tomoko Takayama for their support in laboratory procedures.

## Author Contributions

**Conceptualization:** Azumi Ishizaki, Hiroshi Ichimura.

**Data curation:** Azumi Ishizaki, Tomomi Maeno, Akinori Hara, Sanae Kuramoto, Koichi Nishi, Hiroshi Ichimura.

**Formal analysis:** Azumi Ishizaki, Xiuqiong Bi.

**Funding acquisition:** Azumi Ishizaki, Hiroyuki Nakamura, Hiroshi Ichimura.

**Investigation:** Azumi Ishizaki, Xiuqiong Bi, Quynh Thi Nguyen, Hiroyasu Ooe, Hiroshi Ichimura.

**Methodology:** Xiuqiong Bi, Quynh Thi Nguyen, Hiroyasu Ooe.

**Project administration:** Azumi Ishizaki, Akinori Hara.

**Resources:** Quynh Thi Nguyen, Tomomi Maeno, Akinori Hara, Hiroyuki Nakamura, Sanae Kuramoto, Koichi Nishi, Hiroshi Ichimura.

**Supervision:** Hiroyuki Nakamura, Hiroshi Ichimura.

**Validation:** Xiuqiong Bi, Hiroyasu Ooe, Hiroshi Ichimura.

**Visualization:** Azumi Ishizaki, Xiuqiong Bi.

**Writing – original draft:** Azumi Ishizaki.

**Writing – review & editing:** Xiuqiong Bi, Quynh Thi Nguyen, Tomomi Maeno, Akinori Hara, Hiroyuki Nakamura, Sanae Kuramoto, Koichi Nishi, Hiroyasu Ooe, Hiroshi Ichimura.

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
