## [Decision Letter · Decision Letter 0]

12 Jul 2022

PONE-D-21-40198Neutralizing-antibody response to SARS-CoV-2 for 12 months after the COVID-19 workplace outbreaks in JapanPLOS ONE

Dear Dr. Ichimura,

Thank you for submitting your manuscript to PLOS ONE. After careful consideration, we feel that it has merit but does not fully meet PLOS ONE’s publication criteria as it currently stands. Therefore, we invite you to submit a revised version of the manuscript that addresses the points raised below during the review process.

This must include vaccination status of subjects and also comments on the limitations of this study.

We look forward to receiving your revised manuscript.

Kind regards,

Ray Borrow, Ph.D., FRCPath

Academic Editor

PLOS ONE

Reviewers' comments:

Reviewer's Responses to Questions

**Comments to the Author**

1. Is the manuscript technically sound, and do the data support the conclusions?

Reviewer #1: Yes

2. Has the statistical analysis been performed appropriately and rigorously? 

Reviewer #1: Yes

3. Have the authors made all data underlying the findings in their manuscript fully available?

Reviewer #1: No

4. Is the manuscript presented in an intelligible fashion and written in standard English?

Reviewer #1: Yes

5. Review Comments to the Author

Reviewer #1: The authors examine a very interesting topic with the current study. The results can potentially be useful in the management of the pandemic and the vaccination program worldwide.

The study has several limitation few of which are recognized by the authors:

1. Indeed the sample is rather small (33 subjects) and no definite conclusion can be drawn from these results.

2. No information is given about the vaccination status of these individual. It is very important that the authors include this data in their analysis. Vaccination status, duration until analysis as well as vaccine type are very important information that need to be considered.

3. The authors suggest based on their results that a booster shoot within 1 year will be beneficial given the decline of the Nabs. It will be good if the study included at least few subjects that received a booster shoot in order to validate this point.

4. Absence of re-infection by PCR should have been determined at every stage of the analysis.

6. PLOS authors have the option to publish the peer review history of their article (what does this mean?). If published, this will include your full peer review and any attached files.

Reviewer #1: **Yes: **Kyriacos Felekkis

---

## [Author Response · Author response to Decision Letter 0]

5 Aug 2022

PONE-D-21-40198

Neutralizing-antibody response to SARS-CoV-2 for 12 months after the COVID-19 workplace outbreaks in Japan

Responses to the Editor and Reviewers' comments:

The authors would like to thank the editor and the reviewer for the valuable suggestions and precise comments to clarify the major contribution of the work. Moreover, we sincerely appreciate their great efforts to point out the existing inconsistencies and errors for the improvement. To our best, the manuscript has been carefully revised and clarified according to the editor and reviewer’s comments.

Editor’s comment:

This must include vaccination status of subjects and also comments on the limitations of this study.

 Response: 

Thank you very much for your comment.

Since the COVID-19 vaccination program for general population in Japan began in June 2021, none of the participants had the opportunity to receive COVID-19 vaccination during the study period, from June 2020 to April 2021.

This information has been added in the results section (1) and in the discussion section (2) in the revised manuscript as follows:

(1) None of the participants had the opportunity to receive COVID-19 vaccination and exhibited any clinical symptom suggestive of COVID-19 during the study period. (lines 143–144).

(2) Second, none of the study participants had the opportunity to receive COVID-19 vaccination during the study period, from June 2020 to April 2021, since the COVID-19 vaccination program for the general population in Japan began in June 2021. Recently, however, it was reported that 58 individuals with SARS-CoV-2 infection history in the same area as the present study, including 19 present study participants, had a median anti-RBD-Ab titer of 4.2 log10 U/mL [range 2.8 to 5.0] and Nabs titers of more than 95% after two doses of the vaccines [33]. This recent report would support our suggestion that a booster vaccination within one year after the primary infection will be beneficial, considering that a median anti-RBD-Ab titer of 2.2 log10 U/mL (IQR: 2.0–2.5) and a median Nabs titer of 56.2% (IQR: 39.6–74.4) at 12M in the present study. (lines 334–344) 

Reviewer #1: The authors examine a very interesting topic with the current study. The results can potentially be useful in the management of the pandemic and the vaccination program worldwide. The study has several limitation few of which are recognized by the authors:

Comment 1:

Indeed the sample is rather small (33 subjects) and no definite conclusion can be drawn from these results.

Response:

As reviewer pointed out, the sample number in this study is relatively small to draw any concreate conclusion, and we have already mentioned this issue in the discussion section (lines 334–335). 

In April and May 2020, during the first epidemic of COVID-19 in the study area, we had only six outbreaks with 117 reportedly infected individuals in the study area. We could recruit 33 of them from the two workplace outbreaks in this study, and have successfully followed 26 of the 33 study participants for two years after infection with SARS-CoV-2. We are planning to report the changes of their anti-SARS-CoV-2 antibodies more precisely in relation to COVID-19 vaccination status in another paper.

Comment 2:

 No information is given about the vaccination status of these individual. It is very important that the authors include this data in their analysis. Vaccination status, duration until analysis as well as vaccine type are very important information that need to be considered.

Response:

Thank you for your comment. As mentioned in the response to the editor’s comment, none of the study participants had the opportunity to receive COVID-19 vaccination during the study period. This information has been added in the results section (lines 143–144) and in the discussion section (lines 334 –344) in the revised manuscript.

Comment 3:

The authors suggest based on their results that a booster shoot within 1 year will be beneficial given the decline of the Nabs. It will be good if the study included at least few subjects that received a booster shoot in order to validate this point.

Response:

Following the reviewer’s comment, we have added the following sentences in the discussion section (lines 337–344):

“Recently, however, it was reported that 58 individuals with SARS-CoV-2 infection history in the same area of the current study, including 19 of the current study participants, had a median anti-RBD-Ab titer of 4.2 log10 U/mL [range 2.8 to 5.0] and Nabs titers of more than 95% after two doses of the vaccines [33]. This recent report would support our suggestion that a booster vaccination within one year after the primary infection will be beneficial, considering a median anti-RBD-Ab titer of 2.2 log10 U/mL (IQR: 2.0–2.5) and a median Nabs titer of 56.2% (IQR: 39.6–74.4) at 12M in the current study.”

Comment 4:

Absence of re-infection by PCR should have been determined at every stage of the analysis.

Response:

We would agree that the absence of re-infection by PCR should have been determined at every stage of the analysis. So, we have added the following sentence in the discussion section (lines 312–314): 

In this study, we did not conduct RT-PCR tests to detect SARS-CoV-2 RNA during observation period, “since none of the participants exhibited any clinical symptom suggestive of COVID-19 and testing capacity for COVD-19 RT-PCR in Japan was limited during the study period,”

Further comment from the Reviewer regarding the PLOS Data policy.

3. Have the authors made all data underlying the findings in their manuscript fully available?

Response:

All data, the results of antibodies titers, are fully available as a supporting information in the S1 Table Details of the antibody titers.

---

## [Decision Letter · Decision Letter 1]

12 Aug 2022

Neutralizing-antibody response to SARS-CoV-2 for 12 months after the COVID-19 workplace outbreaks in Japan

PONE-D-21-40198R1

Dear Dr. Ichimura,

We’re pleased to inform you that your manuscript has been judged scientifically suitable for publication and will be formally accepted for publication once it meets all outstanding technical requirements.

Kind regards,

Ray Borrow, Ph.D., FRCPath

Academic Editor

PLOS ONE

Additional Editor Comments (optional):

Reviewers' comments:

Reviewer's Responses to Questions

**Comments to the Author**

1. If the authors have adequately addressed your comments raised in a previous round of review and you feel that this manuscript is now acceptable for publication, you may indicate that here to bypass the “Comments to the Author” section, enter your conflict of interest statement in the “Confidential to Editor” section, and submit your "Accept" recommendation.

Reviewer #1: All comments have been addressed

2. Is the manuscript technically sound, and do the data support the conclusions?

Reviewer #1: Yes

3. Has the statistical analysis been performed appropriately and rigorously? 

Reviewer #1: Yes

4. Have the authors made all data underlying the findings in their manuscript fully available?

Reviewer #1: Yes

5. Is the manuscript presented in an intelligible fashion and written in standard English?

Reviewer #1: Yes

6. Review Comments to the Author

Reviewer #1: The authors have addressed all of my concerns by including additional data or additions to the text.

7. PLOS authors have the option to publish the peer review history of their article (what does this mean?). If published, this will include your full peer review and any attached files.

Reviewer #1: **Yes: **Kyriacos Felekkis
